# Detecting the Presence of Malware and Identifying the Type of Cyber Attack Using Deep Learning and VGG-16 Techniques

Abdullah I. A. Alzahrani [1], Manel Ayadi [2], Mashael M. Asiri [3], Amal Al-Rasheed [2] and Amel Ksibi [2,*]

1   Department of Computer Science, Shaqra University, Shaqra 11911, Saudi Arabia
2   Department of Information Systems, College of Computer and Information Sciences,
    Princess Nourah bint Abdulrahman University, Riyadh 11671, Saudi Arabia
3   Department of Computer Science, College of Science & Art at Mahayil, King Khalid University,
    Abha 62529, Saudi Arabia
*   Correspondence: amelksibi@pnu.edu.sa

**Abstract:** malware is malicious software (harmful program files) that targets and damage computers, devices, networks, and servers. Many types of malware exist, including worms, viruses, trojan horses, etc. With the increase in technology and devices every day, malware is significantly propagating more and more on a daily basis. The rapid growth in the number of devices and computers and the rise in technology is directly proportional to the number of malicious attacks—most of these attacks target organizations, customers, companies, etc. The main goal of these attacks is to steal critical data and passwords, blackmail, etc. The propagation of this malware may be performed through emails, infected files, connected peripherals such as flash drives and external disks, and malicious websites. Many types of research in artificial intelligence and machine learning fields have recently been released for malware detection. In this research work, we will focus on detecting malware using deep learning. We worked on a dataset that consisted of 8970 malware and 1000 non-malware (benign) executable files. The malware files were divided into five types in the dataset: Locker, Mediyes, Winwebsec, Zeroaccess, and Zbot. Those executable files were pre-processed and converted from raw data into images of size 224 * 224 * 3. This paper proposes a multi-stage architecture consisting of two modified VGG-19 models. The first model objective is to identify whether the input file is malicious or not, while the second model objective is to identify the type of malware if the file is detected as malware by the first model. The two models were trained on 80% of the data and tested on the remaining 20%. The first stage of the VGG-19 model achieved 99% accuracy on the testing set. The second stage using the VGG-19 model was responsible for detecting the type of malware (five different types in our dataset) and achieved an accuracy of 98.2% on the testing set.

**Keywords:** malware detection; cybersecurity; machine learning; artificial intelligence; deep learning; VGG-19

## 1. Introduction

Malware is malicious software penetrating different software and data without user authorization [1]. The malicious software targets and infects individual computers or an organization's network. A malware infection has terrible consequences, such as stealing passwords, data theft, blackmailing, etc. The main goal of malware developers is to hurt people and gain access to things that are not open to the public. In 2020, 360,000 new malware files were discovered per day, an increase of 5.2 percent over the previous year. A single infection of malware into an organization's network can lead to the loss of millions of dollars. Many researchers confirmed that the amount of data was doubled every two years, leading to the expansion and growth of malware infections and making it a very serious topic of interest [2].

Malware attacks are also a huge issue that poses a severe threat globally and online. Despite the efforts of researchers and anti-malware companies to reduce malware attacks,

according to a recent cybersecurity threat assessment from Symantec [3], there has been a constant increase in malware. It provides high returns for cybercriminals. Cyber-attackers launch malware campaigns through various channels, including ransomware, banking trojans, viruses, etc. Furthermore, malware is regularly used as a critical attack vector in various cyber-attacks, such as distributed denial-of-service attacks. According to a recent Accenture analysis, the financial cost of a successful malware assault is terrifying, at $2.6 million on average per attack [4]. The constant rise in malware attacks has prompted a flurry of studies into how to reduce malware infections [5–9]. Adopting clever and automatic malware development technologies, such as SpyEye of Zeus and the denial-of-service attacks has enabled this rapid growth in malware manufacturing and delivery [10].

Malware attacks against IoT devices and smart appliances are similar to those directed at regular PCs linked to the Internet. As a result, sophisticated cybersecurity techniques are required to protect millions of IoT users from malicious attacks. Various strategies have been used to identify malware over the years, ranging from intricate hybrid systems to elaborate hand labeling [11].

Machine learning (ML) models have recently been employed in various fields. Moving forward, the most cutting-edge platforms classify malware using image processing and machine-learning or deep learning-based methodologies. The use of machine learning (ML) methods and artificial neural networks (ANNs), which may be coupled into more complicated structures, such as ensemble learning to identify malware from features extracted from malware characteristics, is a typical solution [12–16].

Malware has developed superior capabilities and a wide range of features, increasing the relevance of cybersecurity. Due to the critical importance of the problem above, cybersecurity operations in many businesses have expanded [17,18]. Malware analysis is an essential aspect of cybersecurity. The first step to appropriately protect against malware is discovering harmful software and thoroughly examining its behavior. In this regard, the most critical element is successfully classifying it as harmful software. A family of malicious software likewise represents the malicious conduct to which it belongs. As a result, the actions to take against these activities may differ depending on which malicious software families are involved. Within malware analysis, several processes are usually performed in a row.

Nowadays, deep learning techniques have been proven to obtain solid results in many computer vision tasks and areas and sometimes surpass human intelligence and performance. In this paper, we will take advantage of deep learning in image classification and detection to detect if a given file is malware or benign and specify the type of malware if any. We will convert the benign/malware exe file into a grayscale image. We can treat the task as an image classification task, differentiating between benign grayscale-transformed images and malware ones. In this paper, we made a multi-stage network. The first network decided if the file was malware or benign, and the other specified the malware type if the first network detected the file as malware. This method can help detect malware files early and detect the malware type for resolving the infection.

List of major contributions:

- This work proposed a multi-stage architecture consisting of two modified VGG-19 models.
- Converting benign and malware exe files from raw data into grayscale images.
- Pre-processing techniques were applied to these images.
- The transfer learning approach was applied to our models, which were pre-trained on a Google ImageNet dataset of images of the size 224 * 224 * 3.
- The first stage VGG-19 model achieved an accuracy of 99% on the testing set, and the second stage VGG-19 model achieved an accuracy of 98.2% on the testing set.

The remainder of the paper is structured as outlined below. In Section 2, we discuss previous research undertaken in the same field. The procedure, which comprises the Dataset, pre-processing, model construction, and performance evaluation, is described in Section 3. In Section 4, the results of the tests are presented in the form of graphs, a

confusion matrix, and a discussion of the findings. Finally, the manuscript is ended with the conclusions in Section 5.

## 2. Literature Review

Detecting malicious software with the help of traditional signature and heuristic-based methodologies does not seem to give a high accuracy rate, especially for unknown malware. So, implementing advanced techniques, such as machine learning could help achieve better accuracy rates and solve this problem. Deep learning algorithms and transfer learning techniques improve malware detection resilience and accuracy.

In this work [19], the authors used ResNet-50. They employed RGB pictures of size 224 * 224 with ten folds. For weights, they applied the Glorot uniform technique along with Adam optimization, with a final accuracy of 98.62 percent. The model was trained for 750 epochs. Finally, GIST features with K-nearest neighbors (kNN) where k = 4 were applied, resulting in an accuracy of 97.48 percent and 98.0 percent with bottleneck features.

In another work [20], authors applied two techniques, one ResNet and another GoogleNet. For preparing the data, they used a pipeline and the top model. Further, this model was examined by transfer learning for malware classification. The accuracy of Resnet 18, 34, 50, 101, and 152 was 83 percent, 86.51 percent, 86.62 percent, 85.94 percent, and 87.98 percent, respectively. GoogleNet had an accuracy of 84 percent.

Similar work in [21] employed a VGG16 and ResNet-50 ensemble model. Both networks were fine-tuned. Along with PCA, 90% of the features in the dataset were reduced and fed into a one-vs-all multiclass support vector machine (SVM). They trained their convolutional neural network (CNN) model from 100 to 200 epochs and fine-tuned it for 50 epochs, attaining a 99.50 percent accuracy.

The authors here [22] provided a model with 15 classes and 7087 samples using several feature extraction strategies, with the greatest accuracy of 97.47 percent. Feature extraction techniques, such as GIST descriptors and machine learning algorithms, were applied. It achieved a 97 percent accuracy. Static feature categorization and calculated bi-gram distributions were used in their technique. However, this technique has a fundamental issue: if the opponent is aware of their characteristics, they can take countermeasures and evade detection entirely.

In this work, the authors [23] developed deep learning models combining LSTM hybrid networks and SVM. The accuracy achieved was 77.22 percent for the SVM and CNN models. Another model, which had GRU and SVM, achieved 84.92 percent accuracy. Finally, the model containing the MLP and SVM hybrid model was 80.46 percent, with the particular processing of the pictures.

Similarly, Ref. [24] proposed a CNN–LSTM hybrid model. Their two-layer CNN had a SoftMax and categorical cross entropy and was connected to an LSTM layer with 70 memory blocks and an FCN layer with 25 units. The final accuracy ranged from 96.64 percent to 96.68 percent on various splits.

By extension, the authors [25] implemented two layers—one for 1DD CNN and another for feature extraction with a dropout of 0.1 percent. The memory block for LSTM was 70. They had the highest accuracy rate, at 95.5 percent.

Similar work [26] improved the ResNet5 Model by adding a wholly linked dense layer to the last layer of the model trained on the ImageNet. For malware classification, the SoftMax layer received the output of the fully connected dense layer.

For dynamic picture resampling, the authors [27] proposed using the 'Bat Algorithm' Their goal was to correct the dataset's imbalance. They developed a CNN with 94.5 percent accuracy utilizing this method in addition to data augmentation.

In this work [28], a data equilibrium methodology was proposed based on a genetic algorithm. They achieved an accuracy of 92.1 percent and 96.1 percent correctness with a single objective and 97.1 percent correctness with an algorithm based on multi-objectives.

The authors [29] suggested an ensemble model using extreme learning machines (ELMs) and CNNs. They achieved 96.30 percent accuracy with a single CNN layer and 95.7 percent with two CNN layers.

Another work [30] used a deep learning-based IOT-based hybrid visualization technique. They could create models with accuracies of up to 98.47 percent and 98.79 percent by employing alternative image ratios, although they were reliant on dynamic image attributes.

A mixed methodology with a self-learning system was proposed by [31]. A mixture of CNN BiLSTM and CNN BiGRU models was offered. The accuracy of the proposed models ranged from 94.48 percent to 96.3 percent.

Similar work [32] developed a CNN-based architecture with a byte class, gradient, Hilbert, entropy, and hybrid image transformation with GIST and CNN-based models on the input images. Their GIST produced 94.27 percent accuracy with grayscale images, and CNN performed best with the hybrid image transformation (HIT) technique. Likewise, Moussas et al., suggested a malware detection system that employed both file and picture attributes and was based on a two-level ANN [33]. The first-level ANN employed file features to categorize the malware, whereas the second level of ANNs used malware picture attributes to classify the malware families' confusion.

A new technique was proposed by [34] employing a combination of the first-order and second-order statistical texture features based on the grey-level co-occurrence matrix (GLCM), which were identified using ensemble learning. The Malimg dataset was classified using the kernel-based ELM classifier, which achieved 94.25 percent accuracy.

Another work [35] used the capsule network ensemble model (CapsNet). The CapsNet model uses simple architecture engineering instead of sophisticated CNN architectures and domain-specific feature engineering methodologies. Furthermore, CapsNet does not require transfer learning. Therefore, it is simple to train the model from scratch for Android malware detection.

A robust work [36] suggested combining the developed RNN-LSTM classifier with the NAdam optimization technique. As a result, the accuracy of the performance evaluation on two benchmark datasets was 99 percent.

A malware detection system with high accuracy and flexible resource usage was called MAPAS, according to [37]. MAPAS uses convolution neural networks to assess the behaviors of malicious programs based on their API call graphs (CNN). A lightweight classifier using MAPAS to detect malware can effectively compare API call graphs for harmful activities to API call graphs of applications that will be classified. The evaluation's findings show that MAPAS can categorize applications 145.8% faster than MaMaDroid and utilizes about ten times less RAM. Additionally, MAPAS outperforms MaMaDroid (84.99%) in terms of accuracy in detecting unknown malware (91.27%).

## 3. Materials and Methods

### 3.1. Dataset

The dataset used in this research paper was released in June 2018 [38]. The dataset comprises 8970 malware and 1000 non-malware (benign) executable files. The malware files are divided into five types in the dataset: Locker, Mediyes, Winwebsec, Zeroaccess, and Zbot. Where the malware files are distributed among these mentioned types, there are 300 files of the Locker type, 1450 of the Mediyes type, 4400 of the Winwebsec type, 690 of the Zeroaccess type, and 2100 of the Zbot type. All those malware files are collected from the Malicia dataset [39] and the virus share website. While on the other hand, benign files are collected from the installed files of different legitimate software. All the files are tested and checked by the VirusTotal website to ensure that each file belongs to its correct type.

### 3.2. *Types of Malware*

#### 3.2.1. Locker

Locker is ransomware. After installation on a victim's device, it does not restrict access to files and encrypts them. Instead, it displays a ransom letter on a screen overlay utilizing never-before-seen techniques that use specific Android characteristics to freeze the device.

#### 3.2.2. Mediyes

Mediyes is a Trojan horse or remote access tool dropper program, a type of malicious software that operates similar to a pack mule by downloading and installing additional software on the victim's systems.

#### 3.2.3. Winwebsec

WinWebSec is a rogue security program. WinWebSec scams computer users into buying bogus anti-virus software, similar to most fraudulent security solutions. WinWebSec apps masquerade as anti-virus software but display phony error messages to trick victims into thinking their PCs are infected. ESG malware analysts warn against buying WinWebSec anti-virus software. WinWebSec tools cannot protect your PC from infection and cause chaos.

#### 3.2.4. ZeroAccess

ZeroAccess is a complex scam that generates over 140 million fraudulent ad clicks and 260 terabytes of network traffic daily. The system is named after the company that developed it.

#### 3.2.5. Zbot

The Zeus Trojan, often known as Zbot, is a malware that targets Windows computers to steal confidential financial information. A Zbot can accomplish this through man-in-the-browser assaults (MitB), keyboard tracking (keylogging), and form snatching. Zbots can also carry out assaults using the CryptoLocker malware. Figure 1 shows the distribution of the dataset. We can see that the classes are unbalanced for 1000 files of the benign type and about 9000 of the malware type. Figure 2 shows the virus type distribution.

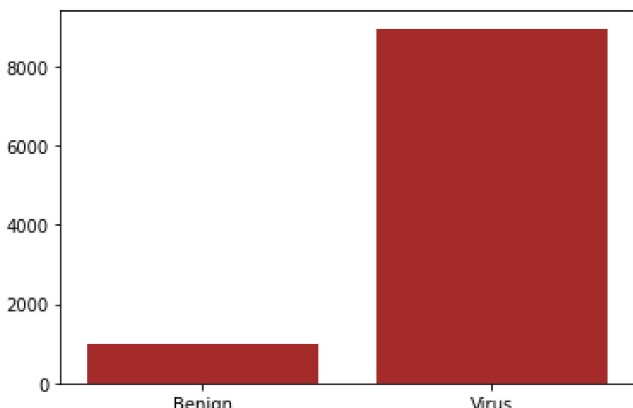

**Figure 1.** Shows the dataset distribution of different virus types in our dataset.

### 3.3. *Pre-Processing*

In this section, we are going to introduce all the pre-processing techniques that we applied on our dataset.

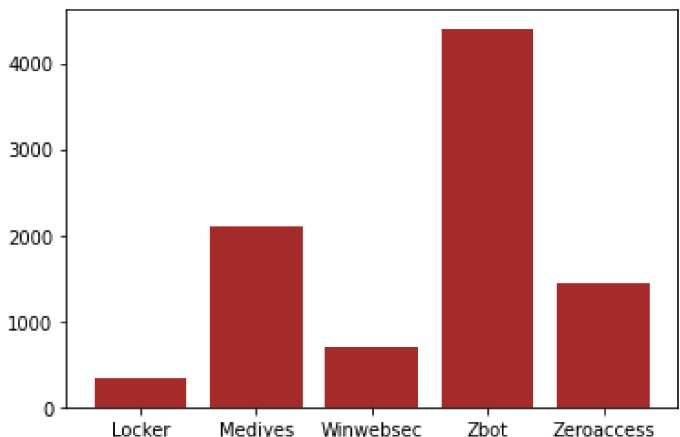

**Figure 2.** Virus type distribution.

### 3.3.1. Converting Benign and Malware Exe Files into Grayscale Images

The main objective of the pre-processing part of our research was to convert the exe files (raw data) in our dataset into images. This is performed by first converting the exe file into a binary representation, then converting this binary application representation into an 8-bit vector, then converting the 8-bit vector to a grayscale image where each pixel value is represented by 8 bits. Figure 3 illustrates the pipeline of converting an exe file into a grayscale image.

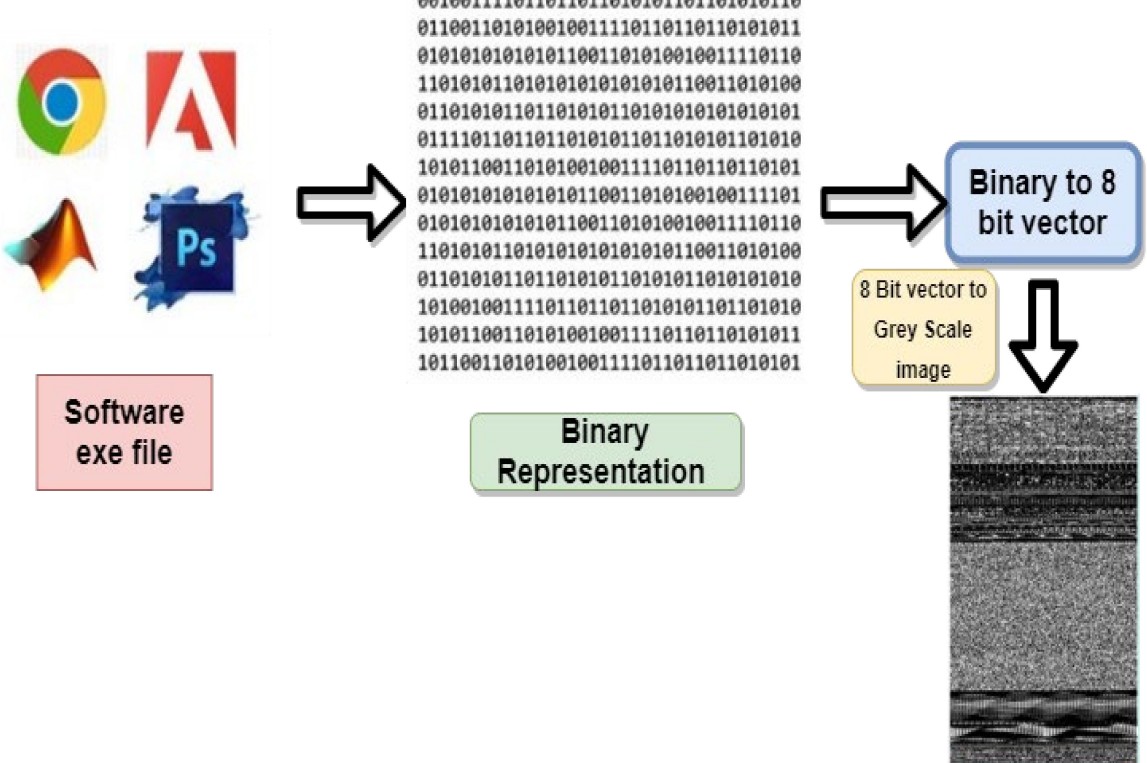

**Figure 3.** Converting exe files to images.

### 3.3.2. Formatting Images

After converting the exe files into grayscale images, those grayscale images are resized to a fixed size of 224 * 224. Additionally, we converted the grayscale images into three-channel images, which are formed by stacking three channels of the grayscale image to

create a single image of three tracks (224 * 224 * 3), as shown in Figure 4 in order to be compatible with applying transfer learning with some models.

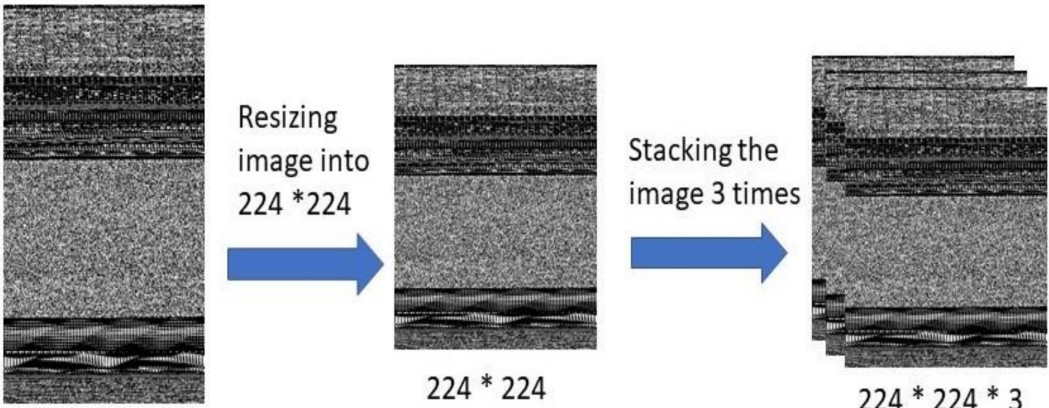

**Figure 4.** Stacking images.

### 3.4. Data Augmentation

We used many data augmentation techniques in our research to balance the number of images in our classes, such as rotating images with different angles, horizontal flipping, vertical flipping, shearing, and zooming.

#### 3.4.1. Rotation

The random rotate enhancement is beneficial because it alters the angles at which items appear in the dataset while you are training.

#### 3.4.2. Horizontal Flipping

The act of flipping is an extension of rotation. It allows the image to be flipped in the up and down direction.

#### 3.4.3. Vertical Flipping

It allows the image to be flipped in the left-right orientation.

#### 3.4.4. Shearing

Shear refers to an axis-based distortion of the image, usually conducted to produce or correct the perception angles. Typically, it enhances the images so that computers can view objects from various perspectives as humans do.

#### 3.4.5. Zooming

The zoom augmentation technique is utilized to magnify an image. This approach zooms the image randomly by either zooming in or adding pixels around the image to enlarge it.

#### 3.4.6. Splitting the Dataset into Training and Testing

We split our dataset into 80% train and 20% test sets. Some of the sample malware images from the different classes are shown in Figure 5.

### 3.5. Methodology

We proposed a multi-stage network in our work. The first stage was responsible for detecting if the given exe file was malware (malicious) or benign. The second stage detected the malware type if the file had been detected as malicious. This section will discuss the methodology used in each stage and the architecture flow.

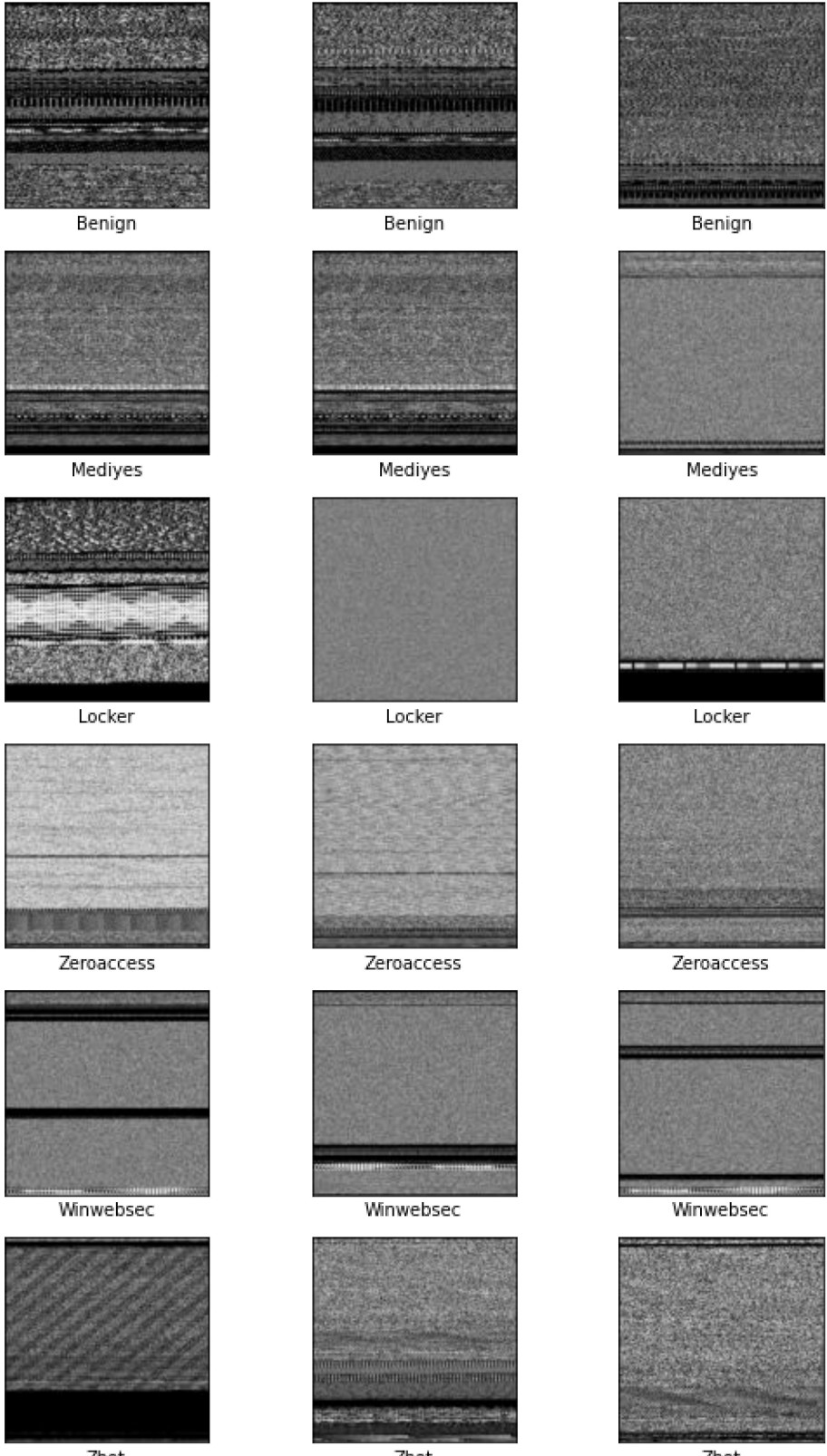

**Figure 5.** Random samples of the classes.

### 3.5.1. Multi-Stage Architecture

First, the exe file is transformed into a grayscale image by converting the file into a binary representation of each 8-bit (byte), which represents a pixel with an intensity from (0 to 255) to form a grayscale image. This obtained grayscale image is resized to a fixed size

of 224 * 224 and stacked three times to create an image of shape 224 * 224 * 3 where we could apply transfer learning to the Google Image-Net dataset [40]. Then, the transformed image of the size 224 * 224 * 3 is fed to the first stage network that is responsible for classifying whether the transformed image is malware or benign; if it is detected from the first stage network as malware, then the image is also fed into a second stage network to detect the type of malware and treat it based on its type. Figure 6 shows the whole proposed methodology that we used in our work. The main architecture used for the two stages was the VGG19 network [41], with some modifications in each stage. The convolutional neural network VGG-19 consisted of 19 layers. It has 16 convolution layers, three fully linked layers, 5 MaxPool layers, and 1 SoftMax layer. Designed as a deep neural network, the VGGNet outperforms benchmarks on numerous tasks and datasets outside ImageNet. In addition, it remains one of the most prominent image recognition architectures.

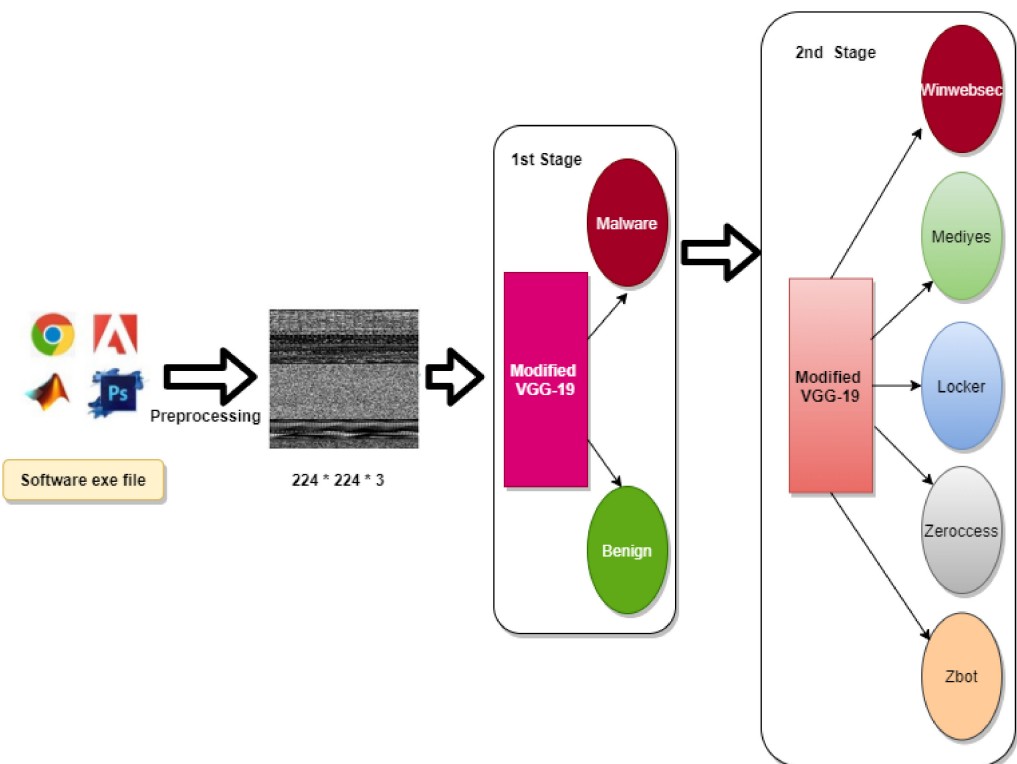

**Figure 6.** Proposed multi-Stage network.

### 3.5.2. First Stage Network

The network used was a modified VGG19 network, as shown in Figure 7. VGG19 is a widely used convolutional neural network due to its simplicity and high performance in image classification and deep learning tasks. VGG19 consists of 19 layers, 16 of which are convolutional layers, and the last three are fully connected layers (dense layers).

Our modified VGG19 network is pre-trained on the Image-Net dataset. The input of the network was 3 channel images of the size 224 * 224 * 3. The image was fed to convolutional layers with a filter size of 3 * 3 for each layer. Those layers were mainly used for extracting features from the input image. The ReLu activation function was applied to each feature map produced after each convolutional layer to add non-linearity to our network. The pooling layers in the network were mainly used for reducing the spatial dimensions of the feature maps by half the size each time. The feature map produced from the last convolutional layer was flattened and fed to fully connected layers. Those layers were used for classification purposes since our problem in the first stage was a binary classification problem. Therefore, the last layer used was the sigmoid layer, which distributes the probability among our two classes, benign and malware, as shown in

Figure 7. If the likelihood of malware was higher, then we fed our image to another stage network responsible for detecting the type of malware.

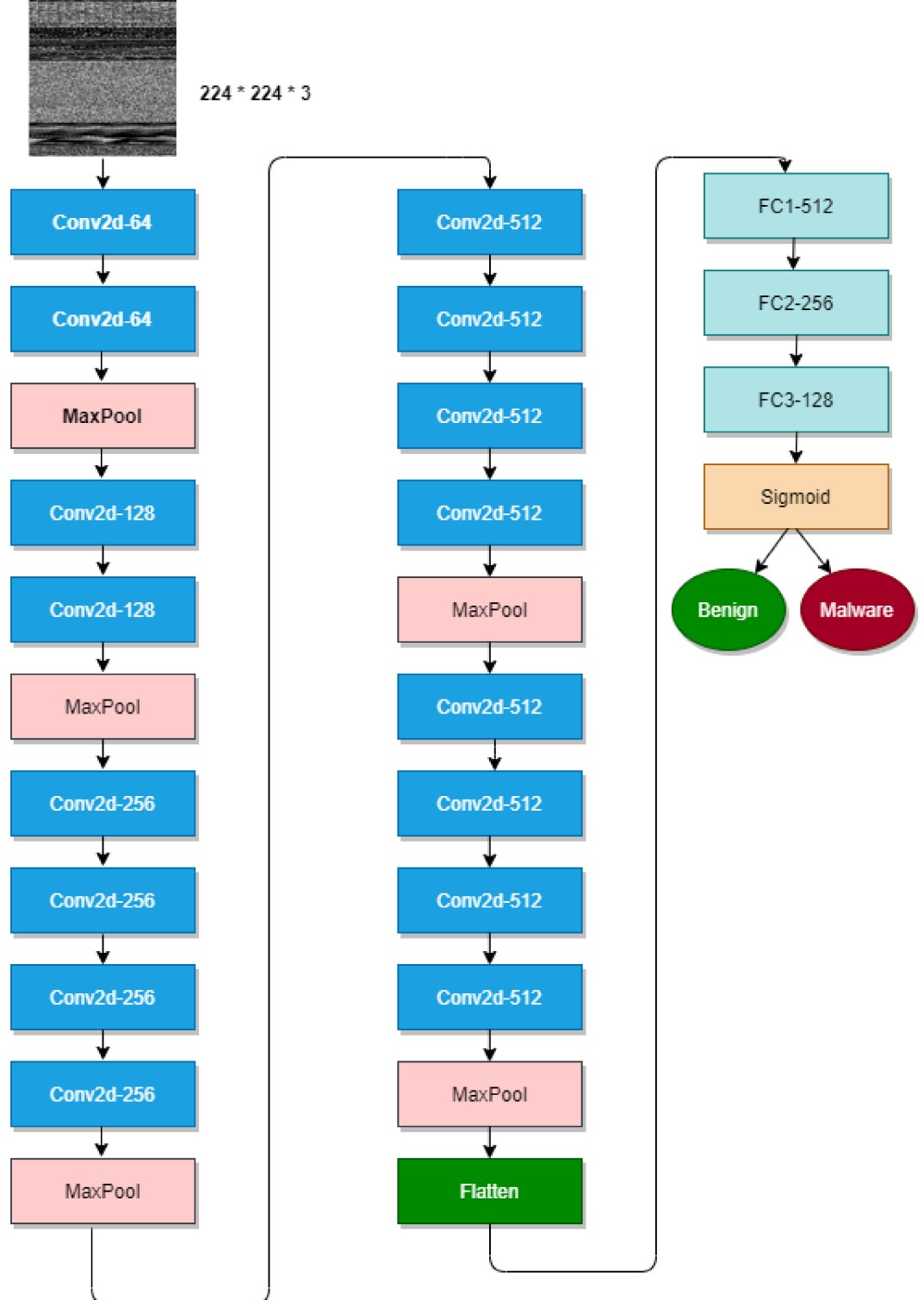

**Figure 7.** First stage VGG19 network.

### 3.5.3. Second Stage Network

As discussed above, the input image was passed through this second stage if the input image was malware. Therefore, the second stage network is responsible for detecting the

type of malware. The used architecture is also VGG19 with other modifications, as shown in Figure 8.

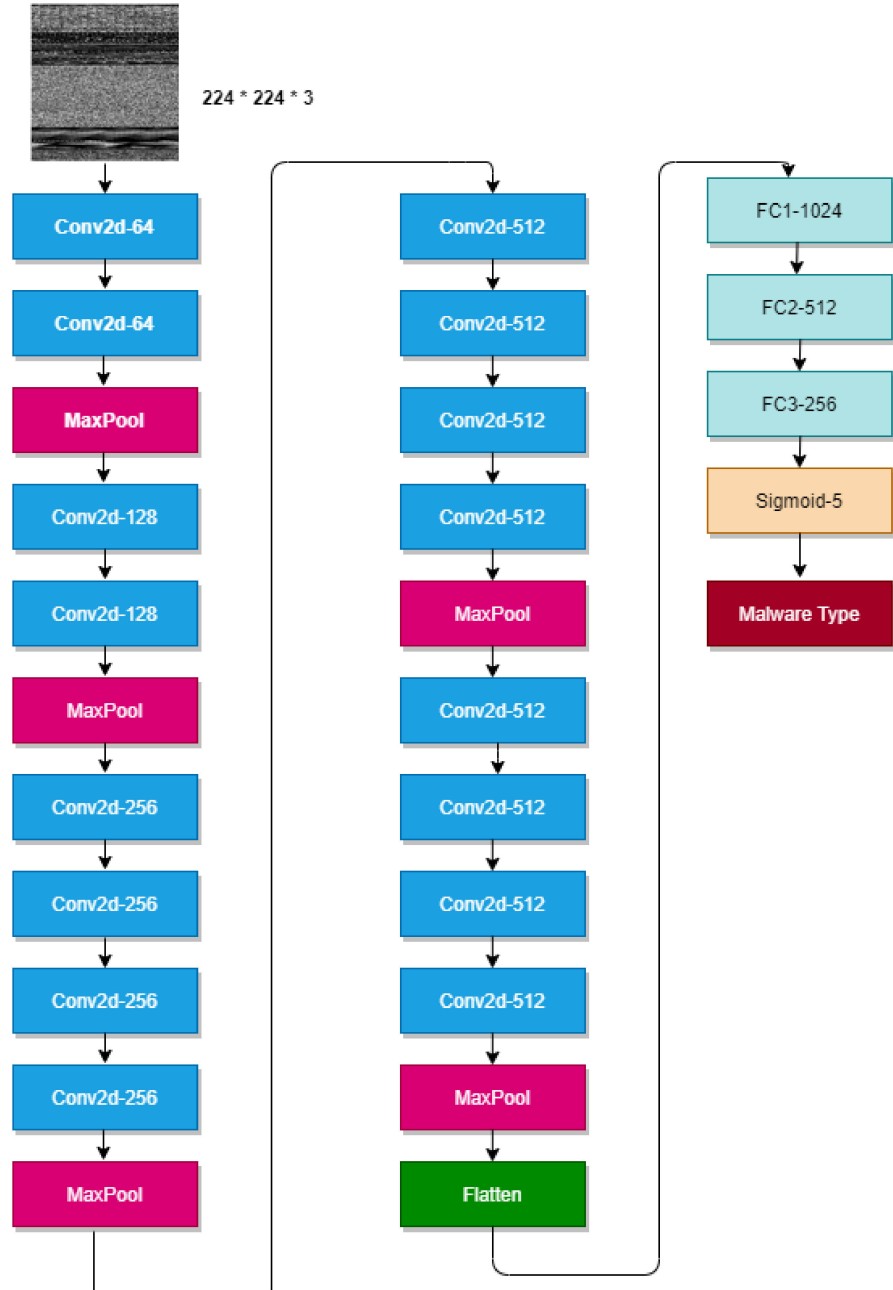

**Figure 8.** Second stage VGG19 network.

The second stage VGG19 network, is also pre-trained on the Image-net dataset. Finally, the last layer used in this network was the SoftMax layer. The activation function known as SoftMax transforms the values of the integers and logits into probabilities. The results of running a SoftMax algorithm are returned as a vector containing each conceivable outcome's probability. The sum of the possibilities includes that the vector always equals one, regardless of the classes or outcomes considered. The SoftMax layer distributed the chances among our 5 different malware classes in the present work.

## 4. Results and Discussion

The first stage of the VGG-19 model was trained for 30 epochs on about 8000 images, and it was evaluated that on about 1600 images, this model was introduced to classify only two classes where the classes were the malware type or benign type. The first stage VGG-19 model achieved an accuracy of 99% on the testing set, while the second stage VGG-19 model was trained for 34 epochs on about 7000 images and was evaluated on about 1500 images, and this Model was trained to classify the different types of malware, which in our case involved five malware classes. The second stage VGG-19 model achieved an accuracy of about 98.2% on the testing set. Figure 9 shows the accuracy of the training vs. the accuracy of the validation at the first stage. The results are for 30 epochs. Figure 10 shows the loss of the training vs. the loss of the validation of the first stage VGG-19 in 30 epochs, while Figure 11 shows the accuracy of the training vs. the accuracy of the validation of the second stage VGG-19 model among the 34 epochs. Further, Figure 12 shows the loss of the training vs. the loss of the validation of the second stage VGG-19 model among the 34 epochs.

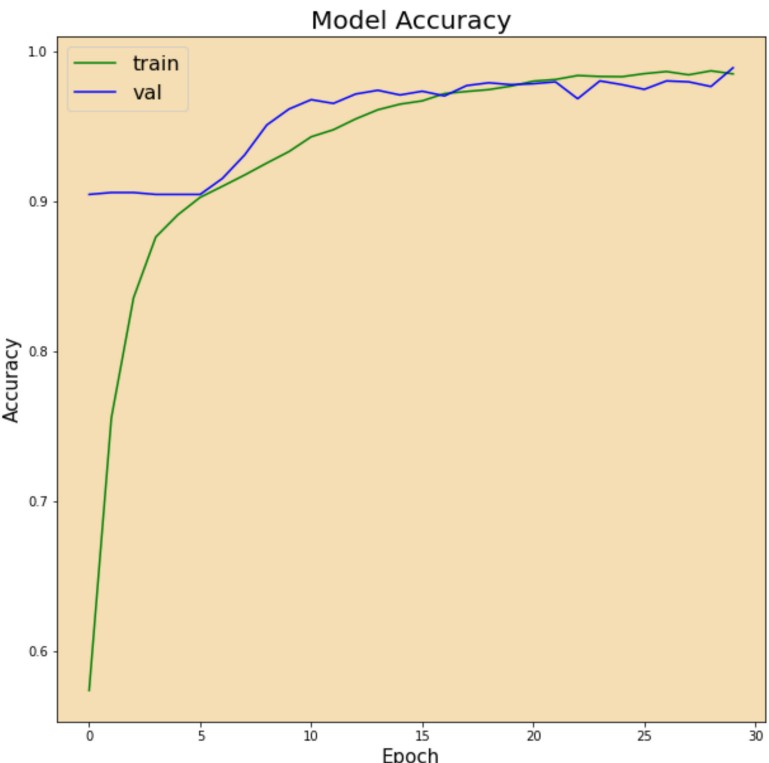

**Figure 9.** Accuracy plot for the first stage VGG-19 model.

We have also used other metrics to measure the model's performance, such as precision, recall, and F1-score, to evaluate our model concerning each class individually. Table 1 shows the first stage of VGG-19 model evaluation on different matrices, while Table 2 shows the second stage of VGG-19 model evaluation on different matrices.

**Table 1.** Model evaluation metrices for the first stage VGG-19 model.

| Class | Precision | Recall | F1-Score |
| --- | --- | --- | --- |
| Benign | 0.96 | 0.94 | 0.95 |
| Malware | 0.99 | 1.00 | 0.99 |

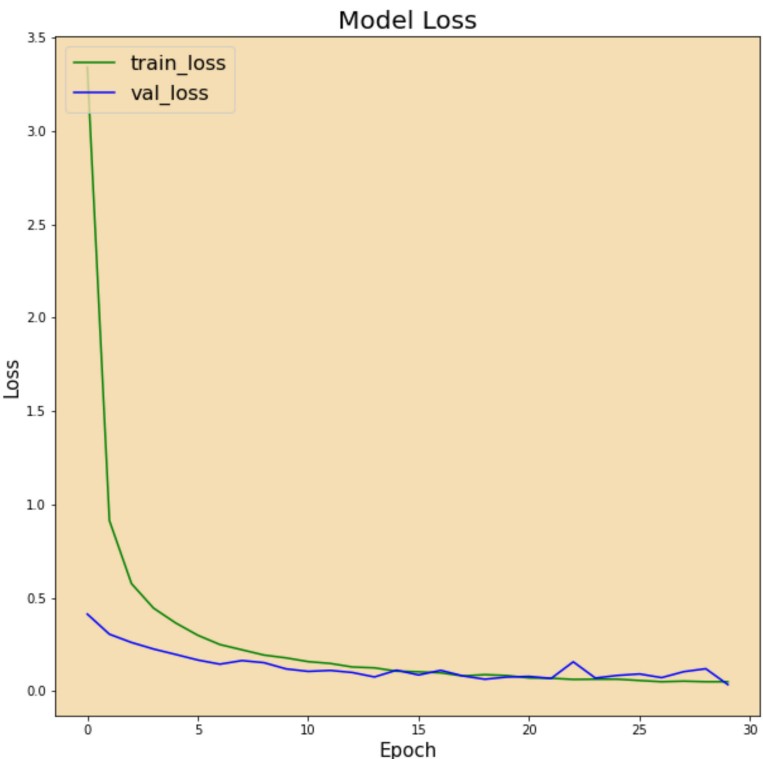

**Figure 10.** Loss plot for the first stage VGG-19 model.

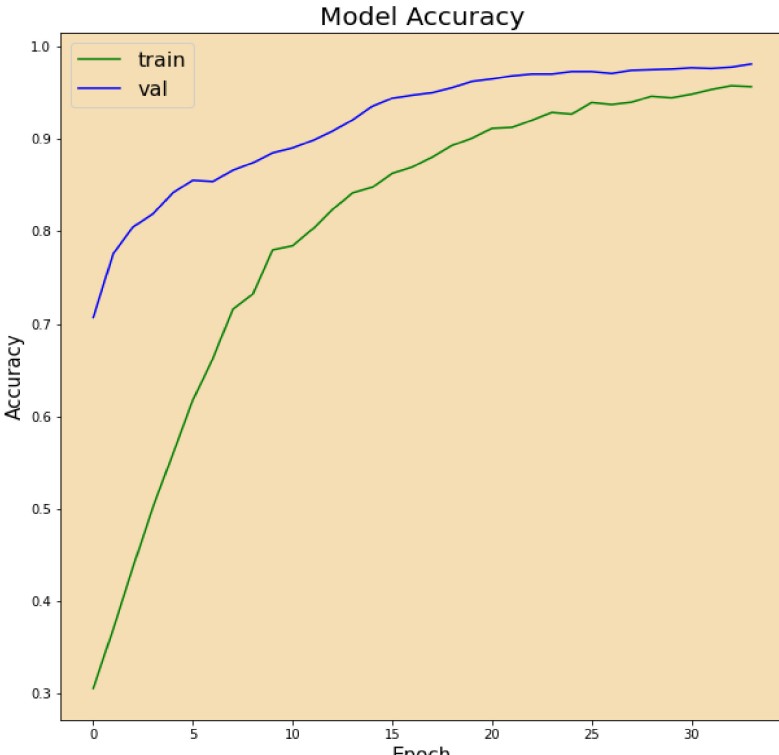

**Figure 11.** Accuracy plot for the second stage VGG-19 model.

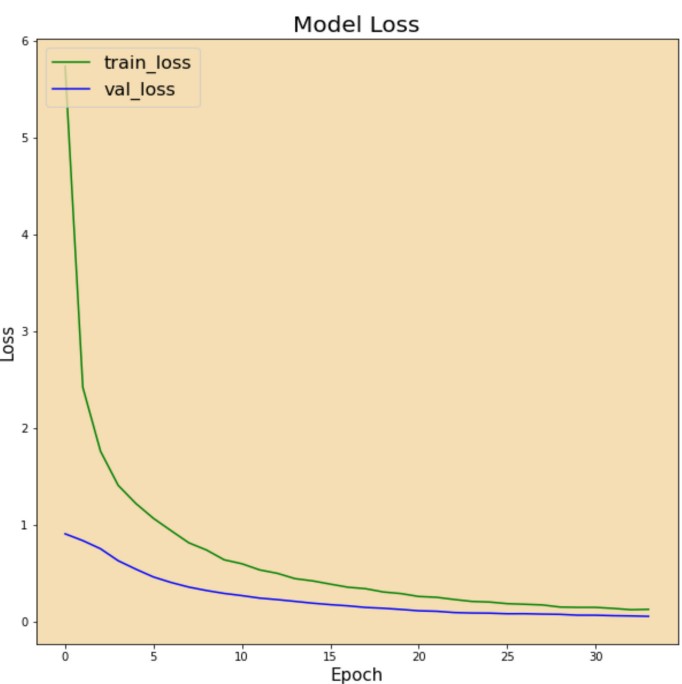

**Figure 12.** Loss plot for the second stage VGG-19 model.

**Table 2.** Model evaluation metrices for the second stage VGG-19 model.

| Class | Precision | Recall | F1-Score |
|---|---|---|---|
| Locker | 0.93 | 0.87 | 0.90 |
| Mediyes | 1.00 | 0.98 | 0.99 |
| Winwebsec | 0.99 | 1.00 | 0.99 |
| Zbot | 0.98 | 0.98 | 0.98 |
| Zeroaccess | 0.93 | 1.00 | 0.96 |

Precision: precision, also known as the positive predictive value, is a term used to describe the accuracy of a prediction. The proportion of the optimistic predictions divided by the total number of positive class values predicted is called precision. Equation (1) shows the equation for calculating the precision metric.

$$Precision = (True\ Positive)/(True\ Positive + False\ Positive) \qquad (1)$$

Recall: it is also called sensitivity. The number of positive class values divided by the fraction of positive predictions equals recall. Equation (2) shows the equation for calculating the recall metric.

$$Recall = (True\ Positive)/(True\ Positive + False\ Negative) \qquad (2)$$

F1-score: the F1-score is also referred to as the F-score or the F-measure. The F1 score represents the balance of precision and recall. The F1-score metric combines the mentioned metrics discussed above (precision and recall) to ensure that the model has high precision and high recall. The value of the F1-score is increased only if the importance of both the precision and recall is high. If the model has a low precision or recall, this will affect the value of the F1 score. F1-score values fall in the interval [0, 1], and the higher the value, the better the classification accuracy [42].

Equation (3) shows the equation for calculating the F1-score metric.

$$F1\text{-}score = (2 * Precision * Recall)/(Precision + Recall) \qquad (3)$$

The two models proposed (the first-stage VGG-19 model and the second-stage VGG-19 model) are evaluated using other matrices, such as the confusion matrix, precision, recall, and F1-score. The confusion matrix is used for evaluation as the accuracy metric can be biased to one of the classes as it measures the model performance on the whole test set, while the confusion matrix measures how well the model performs on each class individually to ensure that the model performs well on all the classes. The confusion matrix is an N*N symmetry matrix where N is the number of classes, and the values in the diagonal of the matrix correspond to the number of correct classifications for each class. Figure 13 shows the confusion matrix of the first stage VGG-19 model, which is 2 * 2 as this model is trained on two classes, while, in Figure 14, the confusion matrix of the second stage VGG-19 model is 5 * 5 as this model is trained on two classes.

**Figure 13.** Confusion matrix of first stage model.

**Figure 14.** Confusion matrix of second stage model.

### 4.1. Discussion

We have used two models in our classification problem. The first stage VGG-19 model is responsible for detecting if the input exe file type is malware or benign. The second stage VGG-19 model is responsible for detecting the type of malware file if the exe file is detected as malware from the first stage model. The first stage VGG-19 model achieved an accuracy of 99% on the testing set. The second stage VGG-19 model achieved an accuracy of about 98.2% As shown in Table 3, the two models were also evaluated using matrices other than the accuracy where these metrics and the results were discussed in detail in this results section.

**Table 3.** Results.

| Method | Testing Accuracy |
| --- | --- |
| First stage modified VGG-19 model | 99% |
| Second stage modified VGG-19 model | 98.2% |

To detect malware before the virus penetrates different software and critical data, many researchers have introduced novel machine learning and deep learning approaches for the early detection of any malware that can lead to disastrous consequences. So many researchers have focused on addressing this problem in order to protect users and companies from infection. In this review section, we will focus our analysis on research that targets malware detection using deep learning and machine learning.

### 4.2. Comparative Analysis

Hemalatha J. et al. (2021) [43] proposed an efficient DenseNet-based deep learning model for malware detection. They worked on many datasets to obtain their results, such as the Malimg dataset, Microsoft BIG 2015, the MaleVis dataset, and the Malicia dataset. They applied different pre-processing techniques to the images, such as converting the binary files into eight-bit unsigned vectors. These vectors correspond to each byte of a pixel with an intensity level value (0 to 255). Further, these byte values are converted to a 2-d array. These arrays are represented and visualized as greyscale images. Finally, those obtained images are resized into fixed sizes of 64 * 64. Then, image resampling techniques are performed by the nearest interpolation method. Those transformed images are split into a ratio of 70 % training set and 30 % testing set, and the training set is fed to a DenseNet model to achieve an accuracy of 98.2% on the Malimg dataset, 98.46% on the BIG2015 dataset, 98.2% on the MaleVis dataset, and 89.48% on the Malicia dataset.

Kumar R. et al. (2018) [44] proposed an approach of malicious code detection based on image processing using deep learning. They worked on three different datasets, which were obtained from different sources. The first two datasets were malicious datasets, the first one was obtained from the Vision Research Lab, and the second one was obtained from the Microsoft Malware Classification Challenge. The third one was obtained by collecting 3000 benign exe files from different sources. They applied different preprocessing techniques to transform the input raw data, such as decompiling the exe file by the IDA algorithm to a binary and assembly and then converting the assembly code to an image, and then those images were resized into a fixed size of 128 * 128 before the images were split into a 90% training set and 10% testing set. The training process was conducted on the training set using a CNN model and achieved a testing set accuracy of 98%.

The study [45] proposed an approach of using deep learning for image-based mobile malware detection. They worked on two datasets: one for Android and one for IOS. They worked on 50,000 Android files (24,553 were malicious among 71 families and 25,447 were non-malicious) and 230 Apple files (115 samples belonged to 10 different families). They applied other pre-processing techniques on the files, such as converting the exe file into a binary file, and then each byte of the binary file was converted to a number from 0 to 255 which then corresponded to a grey scale pixel in the final PNG image. Those transformed images were fed to different machine learning and deep learning models in order

to obtain their results. They obtained an accuracy using a CNN model on the android families of about 92.9%. They obtained an accuracy using a CNN model on the IOS families of about 96.4%.

The study [46] proposed an approach of using visual malware detection by deep learning techniques in a Windows system. They worked on a dataset called Malimg, which consisted of 25 families and 9339 samples. They converted the exe files to greyscale images by converting the file to a binary representation, then to an eight-bit vector, and finally from the eight-bit vector to a greyscale image. Then, those obtained images were resized to 64 * 64 pixels. Then, the images were fed to the CNN model while the last layer consisted of 25 neurons corresponding to one of the Malimg dataset families to achieve an accuracy equal to 96.76%. We reviewed several recent studies that primarily focused on malware detection using deep learning and machine learning approaches; some of these studies are shown in Table 4.

**Table 4.** Comparison of related work.

| Reference | Dataset | Feature Extraction Classification | Accuracy |
|---|---|---|---|
| [43] | They worked on many datasets for obtaining their results, such as Malimg dataset, Microsoft BIG 2015, MaleVis dataset and Malicia dataset. | -DenseNet model | They achieved an accuracy of 98.2% on Malimg dataset, 98.46% on BIG2015 dataset, 98.2% on MaleVis dataset and 89.48% on the Malicia dataset. |
| [44] | The first two datasets were malicious datasets; the first one was obtained from Vision Research Lab and the second one was obtained from Microsoft malware Classification Challenge. The third one was obtained by collecting 3000 benign exe files from different sources. | -CNN Model | 98% |
| [45] | They worked on 50,000 Android file (24,553 were malicious among 71 families and 25,447 were non-malicious) and 230 Apple files (115 sample belonged to 10 different families). | -CNN model | They obtained an accuracy on android families of about 92.9% and on IOS families of about 96.4%. |
| [47] | They worked on a dataset called Malimg which consisted of 25 families and 9339 samples. | -CNN model | 96.76%. |
| Our Research study | The dataset consisted of 8970 malware and 1000 non malware (benign) excutable files. The malware files were divided into 5 different types in the dataset which werw: Locker, Mediyes, Winwebsec, Zeroaccess, Zbot. All those malware files were collected from Malicia dataset and virus share website. | -VGG-19 Model | An accuracy of 99% on the first stage modified VGG-19 model and 98.2% on the second stage modified VGG-19 model. |

## 5. Conclusions

In this work, malware was detected using deep learning techniques. We applied many different pre-processing techniques to the dataset. Benign and malware exe files from raw data were converted into grayscale images. The conversion process was performed by first converting the exe file into a binary representation, then converting this binary application representation into an eight-bit vector, then converting the eight-bit vector into a grayscale image where each pixel value was represented by eight bits. We also applied data augmentation techniques, such as rotating images with different angles, horizontal flipping, vertical flipping, shearing, and zooming. The methodology used in our work was multi-stage architecture. After the raw data were processed and converted into images, these images were fed to this multi-stage architecture where the images were entered first into the first stage VGG-19 model in order to know if the given processed image was malware or benign and if the image was detected as malware from the first stage VGG-19 model, it then entered into the second stage VGG-19 model in order to detect the specific type of malware family (five classes in our case). Precision, recall, and F1-score were evaluated

in order to make sure that our models performed well in each class independently. These metrics were discussed in detail in the results section.

**Author Contributions:** Conceptualization, methodology, writing—original draft, results analysis, A.I.A.A.; data collection, data analysis, writing—review and editing, results analysis, A.K.; methodology, results analysis, M.M.A.; methodology, writing—review and editing, design and presentation, references, A.A.-R.; methodology, writing—review and editing, M.A. All authors have read and agreed to the published version of the manuscript.

**Funding:** The Princess Nourah bint Abdulrahman University Researchers Supporting Project number (PNURSP2022R235), the Princess Nourah bint Abdulrahman University, Riyadh, Saudi Arabia.

**Data Availability Statement:** https://figshare.com/articles/dataset/Malware_Detection_PE-Based_Analysis_Using_Deep_Learning_Algorithm_Dataset/6635642/1.

**Conflicts of Interest:** The authors declare no conflict of interest.

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
