# Peer review of "Detecting the Presence of Malware and Identifying the Type of Cyber Attack Using Deep Learning and VGG-16 Techniques"

_electronics, doi:10.3390/electronics11223665_

Round 1
Reviewer 1 Report
The article presented for review is valuable in terms of cognition, both for theoreticians and practitioners dealing with the issues in question. My doubts are raised by basing the research on data from 4 years ago. With regard to information technology, this period can be compared to a decade or even a century in, for example, the humanities. I believe that the authors should analyze the topicality of the analyzes carried out with the current state. The editorial and language side is also a significant drawback. The article is not carefully prepared and the identified errors affect its overall perception.
Author Response
The authors would like to thank the reviewer for the valuable comments. The authors have checked the manuscript very carefully for the research gap and highlighted it in the contribution list section. The language and English proof of the manuscript is done by native English speakers. The technique applied in this study is very recent and especially for this task, it is not applied. So, it is a rare methodology to solve the issues discussed in the manuscript.

Reviewer 2 Report
1. Kindly highlight the novelty of the proposed method.
2. Proof read the entire manuscript once.
3. Check the Quality of the figures, all figures must be in HD. Figures must be in a standard format.
4. Please modify the abstract and conclusion, both look similar.
5. Check the alignment of all tables and their contents. Need some improvements.
6. check the results section, flow is missing, for example, formulas are introduced after results.
7. Rearrange the paper in sections 3 and 4.
8. Types of malware in line 220 and its types, check format issue.
9. Data Augmentation line no 320 and its types, check format issue.
10. Proposed method / Algorithm missing
11. Results and discussion need improvements how two phases implemented?
Please refer to the below,
10.1186/s13677-022-00326-1
10.1186/s13677-022-00326-1
10.1109/TII.2022.3142306
10.1002/ett.4108
Author Response
- Kindly highlight the novelty of the proposed method.
The novelty of the paper is highlighted in the abstract section of the paper and also in the list of contributions.
“This paper proposes a multi-stage architecture consisting of two modified VGG-19 models. The first model objective is to identify whether the input file is malicious or not, while the second model objective is to identify the type of Malware if the file is detected as Malware by the first Model.”
- Proofread the entire manuscript once.
The whole paper is proofread.
- Check the Quality of the figures, all figures must be in HD. Figures must be in a standard format.
The authors would like to thank the reviewers. Now the following Figures 3, 6, 7, and 8 are redrawn.
- Please modify the abstract and conclusion, both look similar.
Both abstract and conclusion sections are read carefully, and repeated sentences are removed. The abstract is more talking about the problem and proposed methodology to solve this problem while the conclusion is discussing the flow of the work.
- Check the alignment of all tables and their contents. Need some improvements.
Thank you very much. All the tables are now aligned properly.
- check the results section, flow is missing, for example, formulas are introduced after results.
Thank you very much. The flow of the text is corrected and updated. All the equations are moved to the correct position before the results.
- Rearrange the paper in sections 3 and 4.
Both sections are rearranged for the subsection. All the subsections are given proper numbering.
- Types of malware in line 220 and its types, check format issue.
Thank you so much for this comment. The format is corrected and highlighted.
- Data Augmentation line no 320 and its types, check format issue.
Thank you so much for this comment. The format is corrected and highlighted.
- The proposed method / Algorithm missing
The proposed methodology is illustrated in Figures no. 6, 7, and 8 with a detailed description of the methodology in sections 3.4.1 , 3.4.2 and 3.4.3.
- Results and discussion need improvements how two phases implemented?
The two phases are discussed in detail in figure
We have used two models in our classification problem. In the first stage VGG-19 Model is responsible for detecting if the input exe file type is Malware or Benign. The second stage VGG-19 Model is responsible for detecting the type of Malware file if the exe file is detected as malware from the first-stage Model. The First stage VGG-19 model has achieved an accuracy of 99% on the testing set. The Second stage VGG-19 model has achieved an accuracy of about 98.2% As shown in Table 3. The two models were also evaluated using other metrics than accuracy where these metrics and results were discussed in detail in this result section.

Reviewer 3 Report
The paper is well written I don't have any major comments
Author Response
Thank you